# Mitochondrial Dysfunction in Advanced Maternal Aged Cumulus Cells: A Possible Link to ATP Synthase Impairment?

**DOI:** 10.3390/biom14030281

**Published:** 2024-02-26

**Authors:** Sandra Almeida-Reis, Alexandra Carvalho, Conceição Dias, Raquel Brito, Rita Silva, Teresa Almeida-Santos, João Ramalho-Santos, Ana Paula Sousa

**Affiliations:** 1University of Coimbra, Institute for Interdisciplinary Research, PhD Programme in Experimental Biology and Biomedicine (PDBEB), 3030-789 Coimbra, Portugal; sreis@cnc.uc.pt; 2University of Coimbra, CNC-UC—Centre for Neuroscience and Cell Biology, 3004-504 Coimbra, Portugal; teresaalmeidasantos@chuc.min-saude.pt (T.A.-S.); jramalho@uc.pt (J.R.-S.); 3University of Coimbra, CIBB—Centre for Innovative Biomedicine and Biotechnology, 3004-504 Coimbra, Portugal; 4Reproductive Medicine Unit, Centro Hospitalar e Universitário de Coimbra, Praceta Professor Mota Pinto, 3004-561 Coimbra, Portugal4342@chuc.min-saude.pt (C.D.); 26466@chuc.min-saude.pt (R.S.); 5CICS-UBI-Health Sciences Research Centre, University of Beira Interior, 6201-001 Covilhã, Portugal; 6University of Coimbra, Faculty of Medicine, Azinhaga de Santa Comba, Celas, 3000-548 Coimbra, Portugal; 7Eugin Coimbra, Rua Filipe Hodart 12, 3000-185 Coimbra, Portugal; 8University of Coimbra, Department of Life Sciences, Calçada Martim de Freitas, 3000-456 Coimbra, Portugal

**Keywords:** maternal aging, cumulus cells, mitochondria, oxidative phosphorylation, ATP synthase

## Abstract

Age-related changes in the mitochondrial status of human cumulus cells (hCCs) impact oocyte quality; however, the relationship between hCC mitochondrial (dys)function and reproductive aging remains poorly understood. This study aimed to establish the interplay between hCC mitochondrial dysfunction and women’s reproductive potential. In this investigation, 266 women were enrolled and categorized into two groups based on their age: a young group (<35 years old) and an advanced maternal age (AMA) group (≥35 years old). Comprehensive analysis of reproductive outcomes was conducted in our population. Various mitochondrial-related parameters were analyzed across distinct subsets. Specifically, mitochondrial membrane potential (∆Ψm) and mitochondrial mass were examined in 53 samples, mtDNA content in 25 samples, protein levels in 23 samples, bioenergetic profiles using an XF24 Extracellular Flux Analyzer in 6 samples, and levels of reactive oxygen species (ROS) and adenosine triphosphate (ATP) in 39 and 43 samples, respectively. In our study, the reproductive potential of AMA women sharply decreased, as expected. Additionally, an impairment in the mitochondrial function of hCCs in older women was observed; however, no differences were found in terms of mitochondrial content. Regarding oxidative phosphorylation, metabolic profiling of hCCs from AMA women indicated a decrease in respiratory capacity, which was correlated with an age-dependent decrease in the ATP synthase (ATP5A1) protein level. However, intracellular ROS and ATP levels did not differ between groups. In conclusion, our study indicates that age-related dysfunction in hCCs is associated with impaired mitochondrial function, and, although further studies are required, ATP synthase could be relevant in this impairment.

## 1. Introduction

The delaying of motherhood, induced by an increase in educational, employment, and career opportunities [1,2], is one of the factors responsible for an age-related decline in fertility in several contexts due to a decrease in oocyte quantity and quality [3,4].

Although very promising, assisted reproductive technologies (ART) are still inefficient in terms of overall application [5], and their outcomes are hardly predictable [5,6]. Although different variables need to be considered to define the success of ART, oocyte quality and selection are considered crucial, limiting embryo overproduction and shortening the time for take-home baby rates [5,7]. Furthermore, oocyte evaluation based on morphological criteria is still a challenge [7,8,9,10], although some non-invasive objective biomarkers of oocyte quality have been described [11,12].

Cumulus cells (CCs) provide somatic support for oogenesis as the interface between the gamete and follicular microenvironment [5,13,14,15]. The CC transcriptomic, proteomic, and metabolic signatures [5,12,16] were considered potentially valuable non-invasive biomarkers for oocyte competence. In physiological terms, one of the most promising parameters to evaluate CC function is mitochondrial activity, since the mitochondrial function of CCs undergoes dynamic changes and age-related modifications, negatively affecting oocyte competence [17] and ultimately compromising embryonic development [18].

Ageing in the follicular microenvironment is a pleiotropic phenomenon, and age-associated alterations in mitochondria number, morphology/ultrastructure [9,19], distribution [20], and functionality, including mitochondrial membrane potential (∆ψm), metabolism, steroidogenesis [9,21,22,23], mtDNA copy number [24], and mtDNA mutations, result in an overall reduced energy state and increased reactive oxygen species (ROS) production [17]. Reversing these effects could contribute towards the development of new therapeutic strategies capable of mitigating age-related oocyte mitochondrial deterioration. However, although this field is very active, no effective biomarkers have been validated, mainly due to the inconsistency in approaches and findings.

Since aging affects oocyte quality [25], the main goal of this work was to establish a correlation between CC mitochondrial function and age-related decline in reproductive capacity and, ultimately, to provide the basis for further mitochondrial-based strategies to improve ART outcomes for advanced-age women.

## 2. Materials and Methods

### 2.1. Ethical Statement and Participants

The study was approved by the ethics committee of the Centro Hospitalar e Universitário de Coimbra (OBS.SF.059/2022 and CHUC 116/12). Written informed consent was obtained from all participants before hCCs were collected at the Reproductive Medicine Unit of CHUC.

This was a single-center prospective study. From May 2021 to March 2023, a total of 266 patients, aged between 19 and 40 years old, underwent ARTs such as intracytoplasmic sperm injection (ICSI), fertility preservation (FP), or oocyte donation (OD) (Figure 1). The patients were divided into two groups according to their age: the young group (aged < 35 years old, *n =* 140) and the AMA group (aged ≥ 35 years old, *n =* 126). The age cut-off of 35 years old was based on evidence that described a female age-related decline in fertility after the age of 35 [2]. A subset of young women without known infertility (control group, aged ˂ 35 years old, *n =* 40) who were undergoing FP or OD was established. Their clinical and functional parameters were collected and subsequently compared with those of the overall young group (Figure 1).

Women with a body mass index (BMI) of <18.5 or >32 kg/m^2^ and chromosomal abnormalities were excluded. Regarding the FP population, women who underwent chemotherapy cycles and those who had undergone hormone replacement therapy were excluded. Repeated cycles were taken into account whenever improvements were introduced in the controlled ovarian stimulation (COS) protocol or there was a change in the studied variable (age). A maximum of three distinct cycles were considered.

Personal and clinical data were obtained from each sample. The reproductive outcomes (total number of oocytes and metaphase II (MII) oocytes, fertilized oocytes, Day 3 embryos and blastocysts, and clinical pregnancy (CP)) were collected and analyzed. In FP patients, only data concerning oocyte quality was obtained.

### 2.2. IVF Procedure and Sample Collection

COS protocols were decided according to patient age, ovarian reserve, endocrine profile, and ovarian response in the previous cycle(s) by a multidisciplinary team. COS protocols are listed in Supplemental Appendix A. Follicular growth was monitored by ultrasound, and ovulation was triggered when at least one follicle reached 18 mm in diameter. Oocytes were retrieved under transvaginal ultrasound guidance 34–36 h after triggering oocyte maturation.

The procedure for ICSI and embryo culture was performed as standard laboratory procedures, using pre-prepared mediums (Origio, CooperSurgical^®^, Ballerup, Denmark). Briefly, retrieved oocytes were washed with Flushing medium (Origio) with 10 IU/mL Heparin and placed for 1–3 h in Universal IVF Medium (Origio) at 6% ${CO}_{2}$ and 5% $O_{2}$ at 37 °C. Semen was analyzed in accordance with World Health Organization (WHO) guidelines [26] and prepared using a standard gradient separation. ICSI cumulase (Origio) was used to denude oocytes from the hCCs. Microinjection was performed on mature oocytes at 400× magnification, following the Reproductive Medicine Unit’s procedure.

Embryos were cultured individually in preequilibrated sequential culture media, Sequential Cleav™ (Origio) and Sequential Blast™ (Origio), under liquid paraffin in droplets at 6% ${CO}_{2}$ and 5% $O_{2}$ at 37 °C. Developing embryos were graded according to the Istanbul Consensus Workshop Guidelines [27]. Day 3 embryos, or blastocysts, were transferred under ultrasound guidance. Only one embryo was transferred per attempt. The remaining blastocysts or mature oocytes, in FP cases, were cryopreserved using Vitrification Media (Kitazato^®^, Fuji Shizuoka, Japan), according to the manufacturer’s protocol. The blood pregnancy test was carried out 14 days after embryo transfer.

### 2.3. Isolation of hCCs

hCCs were obtained by centrifugation: cell suspension was centrifuged at 500× *g* for 10 min, and sediments were resuspended in 1 mL of phosphate buffer solution (PBS; No. 18912-014; Gibco, Thermo Fisher Scientific, Waltham, MA, USA) and then washed twice with PBS. If red blood cells were present in the sample, they were removed using Red Blood Cell Lysis Buffer (No. 11 814 389 001, Roche, Roche Diagnostics, Mannheim, Germany) according to manufacturer protocol. Finally, cells were washed in PBS. If necessary, hCC populations were resuspended in single-cell suspensions, and cellular density was estimated by the trypan blue exclusion method (Sigma-Aldrich, St. Louis, MO, USA) in a Neubauer chamber.

The cells were randomly and blindly assigned for each protocol. For ∆ψm, 53 samples underwent analysis, while 39 samples were assessed for ROS, and 6 individual samples were used for the oxygen consumption rate (OCR) protocols, all requiring fresh (live) cells. Additionally, mtDNA, ATP content, and Western blotting (WB) assays were conducted on frozen hCCs stored at −80 °C. In each protocol, 25, 43, and 23 samples of hCCs were studied, as schematically shown in Figure 1.

### 2.4. Detection of ∆ψm and Mitochondrial Mass

To analyze mitochondrial activity, ∆ψm and mitochondrial mass were monitored in a total of 53 independent samples. This comprised 27 samples from the young group and 26 samples from the AMA group.

Tetramethylrhodamine methyl ester (TMRM-No. T668, Invitrogen, Thermo Fisher Scientific) probe, which enters active mitochondria based on the positive charge in the mitochondrial matrix, was used to evaluate mitochondrial activity. Mitochondrial mass was measured using Mitotracker^®^ Green (no. M7514, Molecular Probes, Eugene, OR, USA) dye, which enters mitochondria regardless of ∆ψm. Therefore, the red/green fluorescence ratio can be measured as a mitochondrial activity outcome.

Briefly, hCCs were incubated with 50 nM TMRM and 100 nM MitoTracker^®^ Green at 37 °C for 30 min. The fluorescence areas of TMRM and MitoTracker^®^ Green were detected under epifluorescent microscopy (Axio Imager.Z2 optical microscope, Zeiss, Munich, Germany) with excitation wavelengths of 552 nm and 490 nm, respectively. All fluorescence signals were captured as .jpg files using a digital camera (AxioCam HRM, Zeiss) connected to the microscope, and images were analyzed by Image J software (version 1.51j, Bethesda, MD, USA).

### 2.5. mtDNA Content Detection

Total DNA was extracted from 25 samples of hCCs (12 from the young group and 13 from the AMA group) with a commercial QIAamp DNA Blood Mini Kit (No. 51106, Qiagen, Hilden, Germany), following the manufacturer’s instructions.

DNA concentration and quality were determined with a NanoDrop 2000 (Thermo Fisher Scientific). The qPCR procedure for mtDNA content quantification was performed as previously described [28]. A pre-amplification step (10 cycles) was performed with Platinum™ Taq DNA Polymerase (No. 10-966-034, Invitrogen Thermo Fisher Scientific) in a C1000^TM^ Thermal Cycler, in accordance with instructions. Then, pre-amplified products were used to analyze the mtDNA content in hCCs by RT-qPCR using a SsoFast EvaGreen Supermix (No. 172-5200, Bio-Rad Laboratories, Hercules, CA, USA); reaction and data analysis were performed according to Bio-Rad instructions. The reaction took place in the 7500 Real-Time PCR system (Applied Biosystems, South San Francisco, CA, USA). Primers sequences are listed in Supplemental Table S2: mtDNA: mitochondrial minor Arc (mtMinArc); nuclear: beta-2 microglobulin (β2M). The relative mtDNA content was calculated using the −∆∆Ct method and normalized to the housekeeping β2M.

### 2.6. Bioenergetic Profile of hCCs

The hCC bioenergetic profile of 6 individual women was studied, comparing the oxygen consumption rate (OCR) of 3 young women with that of 3 AMA women. Characterization of the bioenergetic properties of hCCs was performed using an XF24 Extracellular Flux Analyzer, and all reagents and plate sets were acquired from Agilent Technologies (Agilent Technologies, Santa Clara, CA, USA), and the assays were run in accordance with manufacturer instructions. This assay monitors key parameters of mitochondrial function by directly measuring the OCR after the addiction of cellular respiration modulators. A total of 5 × 10^4^ single cells were plated in their specific media (DMEM-F12, Gibco Invitrogen; supplemented with 10% FBS, 1/1000 pen/strep), and the assay was performed in XF seahorse DMEM medium supplemented with 25 mM glucose, 4 mM glutamine, and 5 mM pyruvate, pH 7.4. The optimal doses for each mitochondrial modulator were based on manufacturers’ instructions and prior literature [19,29,30]: 1 μM oligomycin, 2 μM FCCP, and 1 μM Rot/AA (each), as these elicited an appropriate response in hCCs.

After OCR analysis, cells were lysed in RIPA lysis buffer (No. R0278, Sigma-Aldrich) supplemented with 2 mM PMSF (phenylmethylsulphonyl fluoride, Sigma-Aldrich), 2× Halt phosphatase inhibitor cocktail (Thermo Fisher Scientific), and protease inhibitor cocktail CLAP (Sigma-Aldrich), according to the manufacturer. Protein concentrations were measured using the bicinchoninic acid (BCA) assay (Thermo Fisher Scientific) following the manufacturer’s protocol, and absorbance was determined in a BioTek Synergy HT multi-detection microplate reader (BioTek Instruments, Winooski, VT, USA) for further normalization of corresponding XF analysis parameters.

### 2.7. Western Blot Analysis of OXPHOS Complexes

A total of 23 hCCs samples (12 from the young group and 11 from the AMA group) were used for OXPHOS protein quantification. hCC protein extracts were isolated using RIPA lysis buffer and estimated by BCA assay. Total protein was separated on SDS-PAGE and immunoblotted. The membranes were probed with primary antibodies against OXPHOS complexes (1:1000; mouse monoclonal cocktail; No. ab110411; Abcam, Cambridge, UK) and β-Calnexin (1:2500, No. ab0041, SICGEN, Cantanhede, Portugal) (used as a loading control) and then visualized by the chemiluminescent detection method using horseradish peroxidase (HRP)-conjugated secondary antibodies specific for primary antibodies. The protein immunoreactive bands were detected using Clarity Western ECL Substrate WesternBright TM Quantum (No. K-12042-D20; Advansta, San Jose, CA, USA) with ImagQuant LAS 500 (GE Healthcare Bio-Sciences AB, Uppsala, Sweden). Densitometric analysis of the band intensity was performed using Quantity One^®^ 1-D analysis software version 4.6.8 (Bio-Rad Laboratories Inc., Hercules, CA, USA).

### 2.8. Cellular ATP Levels

ATP levels were measured using the luciferin-luciferase assay in 43 samples, 24 from the young group and 19 from the AMA group, using a commercial luminescent ATP detection assay kit (No. ab 113849, Abcam) following the manufacturer’s instructions. Standards were prepared and used to draw a standard curve. Then, the cellular ATP levels in hCCs (3 × 10^4^ cells) were measured using a Synergy HT microplate reader. The amount of ATP was calculated using a standard curve. The data were normalized for protein concentration.

### 2.9. ROS Detection

Intracellular ROS levels in hCCs were measured using 2′,7′-dichlorodihydrofluorescein diacetate (H_2_DCF-DA, No. D399, Molecular Probes). hCCs (39 samples; 21 from the young group and 19 from the AMA group) were incubated with 10 μM H_2_DCF-DA at 37 °C for 30 min, protected from the light. The reaction is based on the capacity of cytosolic ROS to oxidize DCFH-DA into a fluorescent compound (DCF). After incubation, hCCs were mounted and visualized. The fluorescence intensity of cytosolic ROS was detected with fluorescence microscopy with excitation wavelengths of 488 nm, and fluorescence signals were captured as .jpg files using a digital camera connected to a fluorescence microscope. Total ROS levels were quantified by analysing the fluorescence intensity of cells using ImageJ software. All photos for analysis were taken using the same intensity parameters and exposure time settings.

### 2.10. Statistically Analysis

The data were analyzed using Prism software (version 8.001, GraphPad, San Diego, CA, USA). The normal distribution was tested, and whenever the data respected the normal distribution, a *t*-test for independent samples was performed. In other cases, the Mann-Whitney test was used to determine statistical significance. A linear regression analysis was performed to identify whether items were associated with age; 95% was the confidence level used for all the analyses. The chi-squared test was used for qualitative variables. For statistical comparison among multiple groups, an ANOVA analysis was applied. To control potential confounding effects derived from sperm quality, all tests applied for ART outcomes were controlled for sperm age and concentration using an analysis of covariance model. The threshold for statistical significance was set at *p* < 0.05. Results indicating statistical significance are denoted as * for *p* < 0.05, ** for *p* < 0.01, and *** for *p* < 0.001. Data are presented as mean ± standard deviation (SD).

## 3. Results

### 3.1. Population Characterization

A total of 266 patients were involved in the study: 140 women in the young group (<35 years old) and 126 in the AMA group (≥35 years old).

To define the relevance of the young group, a subset of young fertile women (considered a control group) without known infertility (FP or OD women) was created. Their clinical and functional parameters were compared with those of the young group. Although the mean age of this group was lower than the younger group, no differences were observed concerning anti-Müllerian hormone (AMH) level and total and MII oocytes, nor for mitochondrial parameters in hCCs (Supplementary Figure S1A–E). Male characteristics were also collected (Supplementary Table S3). Male age and sperm concentration were controlled to ensure that female age was the main relevant variable. Additionally, seminal quality (volume, progressive motility, and motility after swim-up) was similar between groups, although a slightly higher concentration was observed in the AMA group.

The clinical characteristics of both groups are presented in Table 1. The mean age of the young group was 29.74 ± 4.12, while in the AMA group it was 37.11 ± 1.55 (*p* < 0.0001). BMI and social habits were not significantly different.

In terms of possible infertility causes, a higher number of tubal factors in the AMA group was observed, which can be clinically related to a higher rate of secondary infertility. When compared to the young group, the AMA group had lower serum AMH levels and antral follicle counts (AFC). Additionally, fewer total oocytes (9.35 ± 5.53 versus 14.29 ± 8.70; *p* < 0.0001) and MII oocytes (6.24 ± 4.18 versus 8.84 ± 5.70; *p* = 0.0003) were recovered in the AMA group. Fertilized oocytes were also less in the AMA group (3.57 ± 2.80 versus 5.09 ± 3.75; *p* = 0.0027). Regarding embryo development, a reduced number of Day 3 embryos was obtained in the AMA group (2.86 ± 2.47 versus 3.93 ± 3.26; *p* = 0.0103), as well as a lower blastocyst number (2.88 ± 2.49 versus 3.98 ± 3.30; *p* = 0.013). No differences were observed between groups regarding clinical pregnancy rates (*p* = 0.85).

### 3.2. Mitochondrial Activity in hCCs Is Affected by Aging

Mitochondrial membrane potential (∆ψm) was measured using the red/green fluorescence ratio, which reports the population of active mitochondria. In the AMA group, ∆ψm was significantly lower (*p* = 0.0017; Figure 2A,B), suggesting a decrease in the number of mitochondria with appropriate ∆ψm with aging (Figure 2A). Concomitantly, ∆ψm tended to negatively correlate with maternal age (r = −0.26, R^2^ = 0.07, *p* = 0.05; Figure 2C).

### 3.3. Mitochondrial Content Does Not Change in Aged hCCs

To clarify the effects of age on mitochondrial content, mitochondrial mass and mtDNA content were evaluated. Mitochondrial mass showed no significant difference between groups (*p* = 0.81; Figure 2D). On the other hand, the relative mtDNA amount of 25 samples of hCCs from AMA women (*n* = 12) and young women (*n* = 13) was analyzed by RT-qPCR. The results indicate that mtDNA/gDNA were not significantly different between groups (*p* > 0.99; Figure 2E).

### 3.4. Aerobic Respiratory Capacity and OXPHOS Complexes in hCCs Are Affected by Ageing

Age-related modifications in the dynamic metabolic profile of hCCs were evaluated by comparing OCRs, a direct indicator of mitochondrial function. The OCR was monitored using extracellular flux analysis as described in other reproductive cell types [19,29,30]. Assessments were made at baseline (basal condition) and following the sequential addition of mitochondrial function modulators.

As can be seen in Figure 3, respiration in physiological conditions (Figure 3C), corresponding to the real energy demand, is affected by aging (*p* = 0.037). Similarly, the maximum respiration (Figure 3D, *p* = 0.0037) and the spare capacity (Figure 3G, *p* = 0.002) in hCCs were also decreased with age. Furthermore, results seem to suggest an impairment in ATP production (*p* = 0.054; Figure 3E). In fact, our data demonstrated a decrease in oxidative phosphorylation in the AMA group (Figure 3A,C,D,G).

To determine which molecular changes could be involved in this decrease in OCR, we then monitored the content of OXPHOS complex enzymes (complex I–IV and ATP synthase, responsible for electron transfer across the inner mitochondrial membrane and final ATP synthesis, respectively). A significant reduction in ATP5A1 (ATP synthase subunit) protein levels was observed (*p* = 0.046; Figure 4B,C) in hCCs from AMA women. The protein expression of OXPHOS complex subunits showed no significant differences (NDUFB8: *p* = 0.89, SDHB: *p* = 0.75, UQCRC2: *p* = 0.98, MTCO1: *p* = 0.99; Figure 4A,B).

Interestingly, intracellular ATP levels (Figure 5A) in hCCs from the young group were not significantly different from the more advanced age group (*p* = 0.38, Figure 5A). Similarly, no differences were observed in oxidative stress status, which also typically takes place in situations of mitochondrial dysfunction (Figure 5B,C).

## 4. Discussion

Ageing is a complex process characterized by time-dependent deterioration, resulting in a decrease in the quality of life, somewhat mitigated by lifestyle changes and medical advances, allowing for an increase in global lifespan [31].

The female reproductive tract is one of the first to shows hallmarks of ageing [4,32] with an evident decline in the early 30 s [33]. Although ART has been a game changer in recent decades, its success is dependent on oocyte quality. The impact of women’s health condition, age, lifestyle, and environmental factors on oocyte health is well established [25,34]. Thus, as expected, we observed that the reproductive potential of AMA women sharply decreased compared to young women (Table 1). The reproductive potential as well as mitochondrial parameters in hCCs of young women with or without infertility issues were similar, supporting the idea of age-related deterioration of oocyte quality. On the other hand, aged women had significantly fewer retrieved and MII oocytes and a lower number of fertilized oocytes and embryos. These findings align with the concept that infertility in older women may be attributed to a decline in oocyte quality [35]. Our large-scale study, which included a sample that represents the tendency of patients seeking infertility treatments regardless of individual and social characteristics, also stresses that aging is probably the main prevalent factor in ovarian dysfunction.

Mitochondrial impairment has been pointed out as an aging hallmark [3,17,21,31,36], and its involvement in oocyte deterioration in AMA women has already been described [37,38]. Abnormalities in mitochondrial ultrastructure, dynamics and integrity, metabolism, and mtDNA mutations and deletions [17] have been studied in follicular cells.

The crosstalk between CCs and oocytes has an essential role in terms of the oocyte acquiring proper developmental properties. Changes in the mitochondrial function of CCs are potentially a valuable, non-invasive tool to assess oocyte biologic status [5,17]. However, no effective biomarkers have been established, possibly due to difficulty in overcoming the influence of inter-individual characteristics [5]. Our objective was therefore to characterize possible age-related mitochondrial impairment in hCCs.

In fact, an age-dependent decrease ∆ψm in the AMA group (Figure 2A,B) is negatively correlated with maternal age (Figure 2C). These findings are in accordance with previous observations in human granulosa cells [9,19]. Interestingly, the age effect on mitochondrial activity was not dependent on a decrease in mitochondrial mass or mtDNA content (Figure 2D,E). Thus, aging does not seem to affect mitochondrial biogenesis but rather leads to mitochondrial dysfunction, as suggested by the impairment of Δψm, possibly due to structural alterations [39]. Indeed, a previous study, using electron cryotomography, revealed a sequence of age-related events that affect mitochondrial function, including a progressive vesiculation of the mitochondrial inner membrane and a collapse of the cristae that leads to the disassembly of ATP synthase dimers, affecting the capacity to produce ATP [40]. In line with this, we then investigated the OCR profile of hCCs in AMA women (Figure 3). As expected, aged hCCs were compromised, and these cells had a lower respiratory capacity than hCCs from young women. On the other hand, ATP synthase (subunit ATP5A1) expression was decreased in the AMA group (Figure 4B).

These results are in accordance with previous studies that assessed the mitochondrial functions of older women. Liu et al. described an impairment in mitochondrial functions of granulosa cells (GCs) in aged women (aged ≥ 37), related to an OXPHOS dysfunction [9]. Lu and collaborators performed a transcriptomic study of hCCs and found a set of differentially expressed genes, specifically related to OXPHOS, that were downregulated in the AMA group [41,42]. Another study, which investigated the metabolic changes in GCs in patients grouped according to the Poseidon classification, observed that age was the main contributor to OXPHOS dysfunction [19]. In parallel with this mitochondrial dysfunction, a decrease in energy production was previously described [9]. However, in our study, the ATP levels were unchanged (Figure 5A), although a compensatory mechanism can be responsible for maintaining oocyte energy balance to sustain crucial events [43,44,45].

Excessive ROS production is frequently associated with OXPHOS dysfunction [46]. However, our data shows similar ROS generation in both groups (Figure 5B), possibly due to the tight regulation of antioxidant systems [19]. Similarly, others have also described a lack of differences in oxidative stress status in oocyte and follicular cells in young and AMA women [9,47].

## 5. Conclusions

Although there are limitations due to the number of samples in some assays, to our knowledge, this is the first study to identify a specific molecular target, ATP synthase, that seems to be correlated with the mitochondrial dysfunction of hCCs in reproductive aging. Taken together, and although more studies are needed, our data suggests a pattern of mitochondrial abnormalities in hCC, mainly associated with ATP synthase, which could compromise oocyte quality and be associated with poor ART outcomes. This suggests the possibility of the development of targeted therapeutic strategies that may mitigate the effects of oocyte aging, such as enhancing or boosting mitochondrial function in hCCs.

## Figures and Tables

**Figure 1 biomolecules-14-00281-f001:**
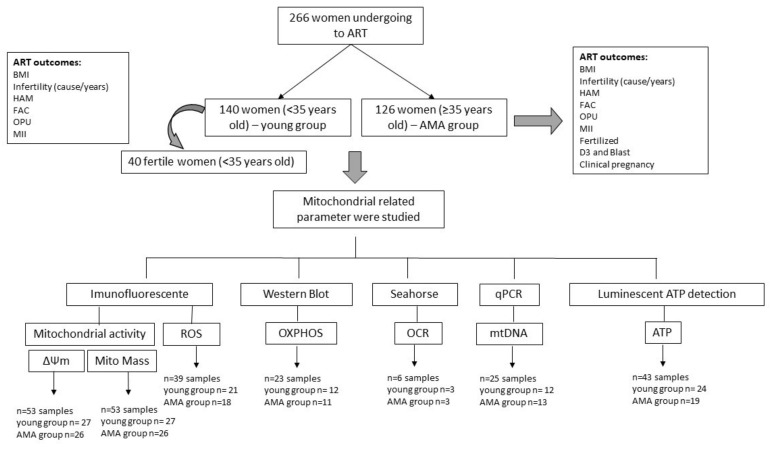
Schematic representation of the experimental design and samples analyzed. A total of 266 women were enrolled in the study and divided into two groups according to the age cut-off (35 years old). To ensure that age was the main relevant factor, a subgroup of fertile young women was created (*n =* 40) and compared with the young group. To analyse mitochondrial parameters, the samples were allocated into different groups: ∆Ψm (*n =* 53), mitochondrial mass (*n =* 53), ROS levels (*n =* 39), OXPHOS protein levels (*n* = 23), bioenergetic prolife (*n* = 6), mtDNA content (*n* = 25), and ATP levels (*n* = 43).

**Figure 2 biomolecules-14-00281-f002:**
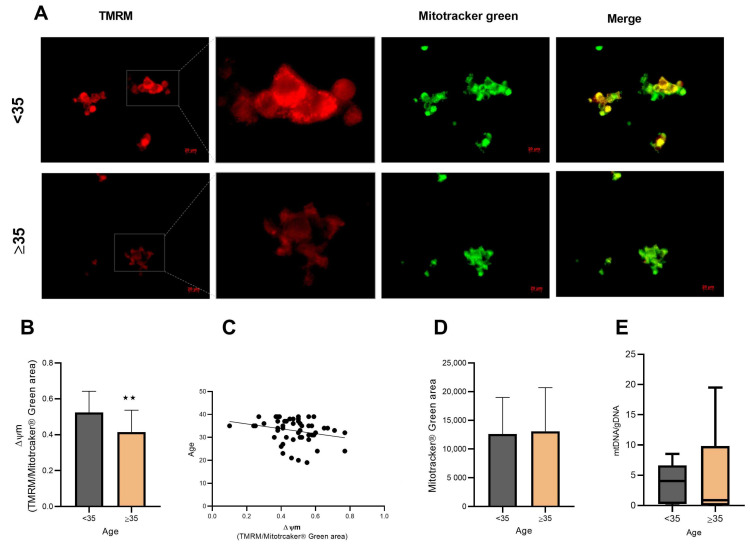
Mitochondrial function was impaired in hCCs from older women. TMRM (50 nM) and Mitotracker^®^ Green (100 nM) fluorescence area were monitored under fluorescence microscopy at 525 nm and 488 nm to indicate Δψm and mitochondrial content, respectively. (**A**) Representative fluorescence images of TMRM and Mitotracker^®^ Green among groups. Scale bar 20 μm (**B**) Median fluorescence area of cells positive TMRM for each group. Results are presented as mean ± SD (<35 *n* = 27 and ≥35 *n* = 26). A decrease in Δψm (*p* = 0.0017) in hCCs from the AMA group was observed. (**C**) Regression analysis of Δψm in the hCCs with age of patients (*n* = 53). The linear correlation between TMRM fluorescence in hCCs and age was analyzed, and Δψm tended to negatively correlate with maternal age. The linear equation was Y = −10.59×x + 37.98, with *n* = 53, r = −0.26, R^2^ = 0.07, and *p* = 0.05. (**D**) Mitochondrial mass was detected by the fluorescence area of Mitotracker^®^ Green, and no difference was observed between groups (*p* = 0.81) (<35 *n* = 27 and ≥35 *n* = 26). (**E**) mtDNA content in hCCs from the young group and old group was analyzed by RT-qPCR. The box plot shows that the mtDNA content in hCCs from patients over 35 years old (*n* = 13) was equal (*p* > 0.99) to that of younger patients (*n* = 12). Statistical significance was considered when ** *p* < 0.01. Specific statistical tests: a *t*-test was performed for the ∆ψm and mitochondrial mass; a Mann–Whitney test was performed for mtDNA content.

**Figure 3 biomolecules-14-00281-f003:**
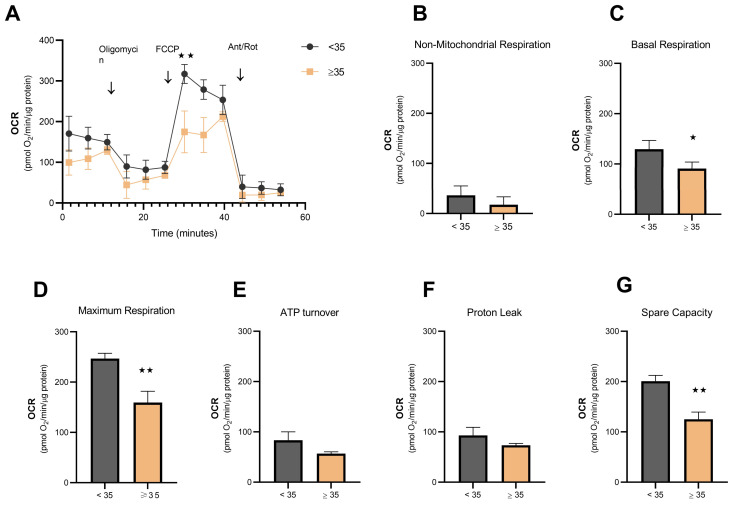
Aerobic respiratory capacity is impaired in aged hCCs. (**A**) The OCR was measured using the Seahorse XFe24 Analyzer in hCCs from young (*n* = 3) and AMA (*n* = 3) women to detect several parameters related to aerobic respiratory capacity, including non-mitochondrial respiration (**B**), basal respiration (**C**), maximum respiration (**D**), ATP turnover (**E**), proton leak (**F**), and spare capacity (**G**). The aerobic respiratory capacity of the aged group was decreased in terms of basal respiration (*p* = 0.037), maximum respiration (*p* = 0.0037), and spare capacity (*p* = 0.002); no differences were observed in non-mitochondrial respiration (*p* = 0.25) or in proton leak (*p* = 0.105). In terms of ATP turnover, data suggest a decrease in ATP production (*p* = 0.054). In each assay, 5 × 10^4^ cells were plated overnight in experimental conditions and three measurements of OCR were performed before and after the sequential injection of each of the four compounds: Oligomycin (1 µM); FCCP (2 µM), and Antimycin A + Rotenone (1 µM each), respectively. Parameters of aerobic respiration were obtained as follows: non-mitochondrial respiration = OCR (AA/Rot); basal respiration = OCR (basal) − OCR (AA/Rot); ATP turnover = OCR (basal) − OCR (oligo); maximum respiration = OCR (FCCP) − OCR (AA/Rot). Spare capacity = OCR (FCCP) − OCR (basal) and Proton leak = OCR (Oligo) − OCR (AA/Rot). The arrow indicates the moment of compound injection. Specific statistical tests: a *t*-test was performed. Results are presented as the mean ± SD of three replicates. Statistical significance was considered when * *p* < 0.05 and ** *p* < 0.01.

**Figure 4 biomolecules-14-00281-f004:**
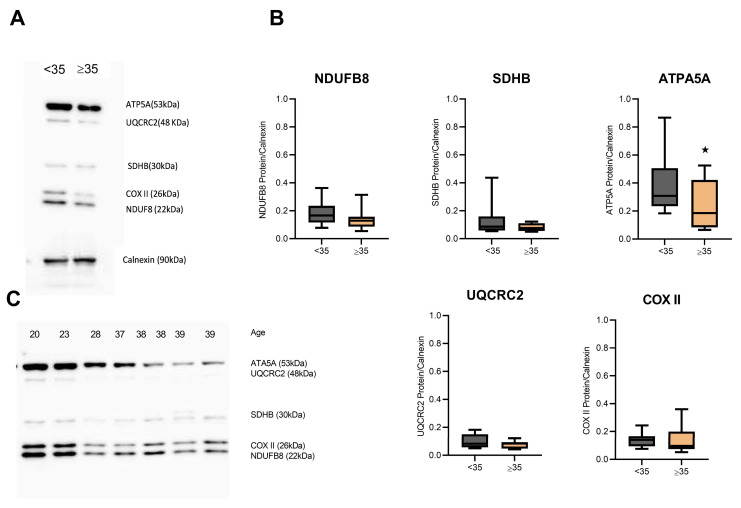
OXPHOS complex levels in aged hCCs. OXPHOS components in hCCs from the young group and the AMA group were monitored by Western Blot (WB). In each assay, 20 μg of protein lysate was separated using a 12% SDS-PAGE, and protein loads were evaluated using calnexin, suggesting that the samples were loaded with a similar amount of protein in each lane. (**A**) Representative analysis of the WB of OXPHOS subunits (Complex I—NDUFB8, Complex II—SDHB, Complex III—UQCRC2, Complex IV—COX2, and ATP synthase—ATP5A) of hCCs in the young and AMA groups. (**B**) Densitometry of the OXPHOS subunit protein detection for young (*n* = 12) and AMA (*n* = 11) groups, relative to calnexin. The expression levels of ATP5A-related proteins were significantly decreased (*p* = 0.046) in the aged group compared with the young group. On the other hand, NDUFB8 (*p* = 0.89), SDHB (*p* = 0.75), UQCRC2 (*p* = 0.98), and COX2 (*p* > 0.99) subunits showed no differences between groups. (**C**) Representative analysis of the WB of OXPHOS subunits of hCCs in three young women and five AMA women. A one-way ANOVA test was performed for WB. Statistical significance was considered when * *p* < 0.05. Original western blotting figures can be found in Appendix A.

**Figure 5 biomolecules-14-00281-f005:**
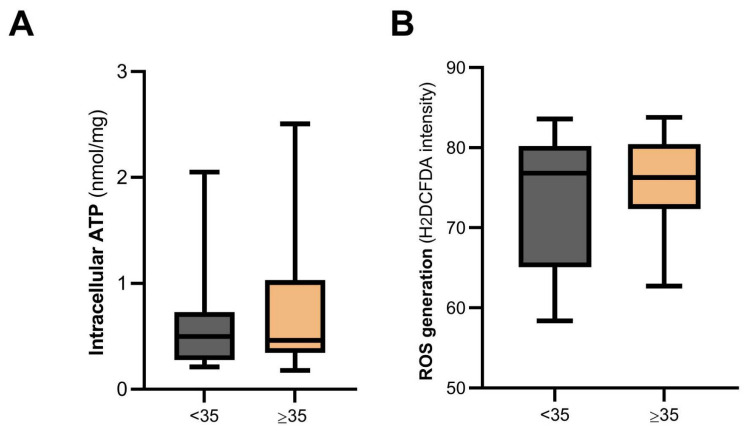
ATP and reactive oxygen species (ROS) levels in hCCs were not affected by aging. (**A**) Intracellular ATP content was measured by a luciferin-luciferase assay in hCCs from the young group (*n* = 24) and AMA group (*n* = 19). The box plot showed that the median intracellular ATP level for hCCs from the young and AMA groups was equal (*p* = 0.48). (**B**,**C**) Intracellular ROS in hCCs of the young group (*n* = 21) and AMA group (*n* = 18) were detected by fluorescence microscopy using H2DCFDA (10 μM), and nuclei were counterstained with Hoechst 33342 (5 μg/mL). (**B**) Representative fluorescence images of H2DCFDA and Hoechst 33342 for each group. (**C**) The boxplot represents the median fluorescence intensity of hCCs in women under 35 years old (*n* = 16) and above 35 years old (*n* = 13). The analysis indicates that there was no statistically significant difference in terms of intracellular ROS levels between the two groups (*p* = 0.79). A Mann–Whitney test was performed for both intracellular ATP and ROS levels. Scale bar: 20 μm.

**Table 1 biomolecules-14-00281-t001:** Clinical characteristics of patients involved in the study.

Characteristics	Young Group	Advanced-Maternal Age Group	*p*-Value	*
Number of patients	140	126		
*Age (y)*	29.74 ±4.12	37.11 ± 1.55	<0.0001	***
*BMI (Kg/m^2^)*	23.73 ± 4.24	24.44 ± 4.02	0.162	NS
*Social Habits (n/%)*				
Smoking	17 (13.71%)	17 (13.49%)	0.85	NS
Alcohol	2 (1.60%)	1 (0.79%)	0.62	NS
*Duration of infertility (Y)*	4.84 ± 1.96	4.98 ± 2.31	0.80	NS
*Primary/secondary infertility*	97.96/2.04	77.24/22.76	0.0001	***
*Infertility Cause (n/%)*				
Male factors	48 (34.29%)	49 (38.89%)	0.447	NS
Fallopian tube factors	25 (17.86%)	45 (34.13%)	0.003	**
Idiopathic	4 (2.86%)	5 (3.97%)	0.739	NS
Ovarian factor	29 (20.71%)	26 (20.63%)	>0.999	NS
No infertility cause	27 (19.29%)	4 (3.17%)	<0.0001	***
*Antral follicle count (AFC)*	16.64 ± 9.10	12.35 ± 7.03	0.0002	***
*Basal serum AMH (ng/mL)*	4.29 ± 3.22	3.09 ± 2.56	0.001	***
*OPU egg number*	14.29 ± 8.70	9.35 ± 5.53	<0.0001	***
	(137)	(127)		
*MII oocyte number*	8.84 ± 5.70	6.24 ± 4.18	0.0003	***
	(127)	(126)		
*Fertilized oocytes*	5.09 ± 3.75	3.57 ± 2.80	0.0027	**
	(101)	(122)		
*Embryos D3*	3.93 ± 3.26	2.86 ± 2.47	0.0103	*
	(101)	(122)		
*Blastocysts*	3.98 ± 3.30	2.88 ± 2.49	0.013	*
	(101)	(122)		
*CP (n/%)*	20 (36.36%)	20 (33.90%)	0.85	NS

Note: Data are presented as the mean ± standard deviation (SD) or number/%, with *p* < 0.05 indicating statistical significance. AFC (basal antral follicle count); AMH (anti-mullerian hormone); BMI (body mass index); CP (clinical pregnancy); D3 (Day 3); MII (metaphase II); OPU (ovum pick-up). NS, equals not significant, * *p* < 0.05, ** *p* < 0.01, and *** *p* < 0.001.

## Data Availability

The data underlying this article are available in the article and in its online supplementary material.

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
