# Peer review of "Mitochondrial Dysfunction in Advanced Maternal Aged Cumulus Cells: A Possible Link to ATP Synthase Impairment?"

_biomolecules, 2024, doi:10.3390/biom14030281_

Round 1
Reviewer 1 Report
Comments and Suggestions for Authors
The submitted manuscript is very interesting and clearly written. The authors are shown monitoring cumulus cell respiration. Although the methods used are at a good level, basically the differences in age may not be so significant as to be reflected in the monitored parameters. However, the authors found some differences that they present.
Major point: 14.29 oocytes were isolated, but only 8.84 matured to the MII stage (in the young category and it was 9.35 and 6.24 in the old category). That's a low number, how many were GV and how many MI? CCs were isolated from all COCs? There may also be oocytes immature or in atresia, this could affect the results. In some methods, only a few samples were analyzed (bioenergetic 3 women)
Please describe the method of hCG sample preparation
Minor point:
abbreviation hCG is not explained
in references Vitalae are 7wo years ...2016, 2016...
Comments on the Quality of English LanguageEnglish Language is good
Author Response
Thank you for your very helpful comments.
Regarding concerns about the number of immature and low-quality oocytes, in general, the young group actually tends to present a higher number of oocytes that are not usable. The number of immature (GV+MI) observed in the young group was 2.38 versus 1.54 in AMA group. While in terms of low-quality oocytes (atretic+ degenerated) the young group had 3.36 versus 1.57 in the AMA. This would suggest that the younger group could show poorer parameters, when in fact we observed the exact opposite.
We accept that the inclusion of CCs from the entire pool of COCs, encompassing mature, immature, and degraded oocytes, could be considered a potential limitation in our study. However, even though more unusable oocytes were found in the younger group, their mitochondrial properties were still globally superior.
Please describe the method of hCG sample preparation
To provide a more comprehensive understanding of our methodology, here's a detailed outline of our sample preparation process, which we have now included in the revised Manuscript:
For each patient, the hCCs, that were collected in Universal IVF Medium, were obtained by centrifugation. The hCC cell suspension was centrifuged at 500g for 10 min, and sediments were resuspended in 1 mL of phosphate buffer solution (PBS; No. 18912-014; Gibco, Thermo Fisher Scientific, Waltham, MA, USA) and then, washed twice with PBS. If red blood cells were present in the sample, red blood cell lysis buffer (No. 11 814 389 001, Roche, Roche Diagnostics, Mannhein, Germany) was added for 10 min at room temperature, then removed by centrifugation at 500g for 10 min. Finally, cells were washed in PBS. Cells were then centrifuged at 500g for 10 min, resuspended in a single-cell suspension, and counted using trypan blue exclusion method (Sigma-Aldrich, St. Louis, MO, USA) in a Neubauer chamber: 3x104 cells were allocated for ATP content assay and 5x104 hCCs/well were recovery for aerobic respiration assay.
Fresh (live) hCCs were required to MMP, ROS, Bioenergetic profile protocols. mtDNA, ATP content and WB assays were performed using frozen hCCs, stored at -80°C until experiment day.
Minor point:
abbreviation hCG is not explained -
in references Vitalae are 7wo years ...2016, 2016...
Both issues were corrected.
Reviewer 2 Report
Comments and Suggestions for Authors
Mitochondrial dysfunction in ovarian ageing is still poorly understood and requires further study. The authors of this study have provided some very interesting new results, particularly on the bioenergetic profile and a decrease in ATP synthase protein in the cumulus cells of older women.
This article represents a lot of work, investment and a conclusion providing new research elements to explore. The authors provide a lot of information about the material and the method, which is very useful. This article is clear, well-constructed, and has no major flaws, so I recommend its publication after some minor clarifications :
-l108 : Please be careful with your reference 26, which does not refer to the WHO manual on sperm treatment. Prefer the following reference: World Health Organization. WHO laboratory manual for the examination and processing of human semen Sixth Edition. 6th ed. Geneva: World Health Organization, 2021.
-The distribution of samples for all the experiments is not very clear. Can you give details in the materials and methods on the number of samples used for Western blotting, cellular ATP levels and ROS detection ? Your conclusion on the robustness of your results as a function of the number of samples is very welcome (l495-499). Our field of research limits us in the number of samples and it's always very relevant to point this out.
-l155 and l319 : Please be careful with the acronym « RT-qPCR ». According to the article « MIQE Guidelines : Minimum Information for Publication of Quantitative Real-Time PCR Experiments » (Bustin et al., 2009). « We propose that the abbreviation qPCR be used for quantitative real-time PCR and that RT-qPCR be used for reverse transcription– qPCR. Applying the abbreviation RT-PCR to qPCR causes confusion and is inconsistent with its use for conventional (legacy) reverse transcription–PCR. »
- Can you explain the choice of relative mtDNA quantification?
The analysis of mtDNA content in absolute quantification is more appropriate and accurate. Moreover, it is impossible to relate your results to other articles in the literature (Ogino et al. 2016, Desquiret-Dumas et al. 2017, Taugourdeau et al. 2019, Yang et al 2021, Liu et al. 2021) describing an absolute quantification of mtDNA in cumulus cells.
I invite the authors to read the article by Li et al (Pharmacol Ther 2021, Emerging methods for and novel insights gained by absolute quantification of mitochondrial DNA copy number and its clinical applications). This review summarises the methods used for mtDNA copy number quantification and in particular the advantages of absolute quantification.
-Mitochondrial primers have been tested on Rho0 cells? This is essential to avoid pseudogenes amplification.
Author Response
-l108 : Please be careful with your reference 26, which does not refer to the WHO manual on sperm treatment. Prefer the following reference: World Health Organization. WHO laboratory manual for the examination and processing of human semen Sixth Edition. 6th ed. Geneva: World Health Organization, 2021.
Thank you for your very helpful comment. The reference was changed.
-The distribution of samples for all the experiments is not very clear. Can you give details in the materials and methods on the number of samples used for Western blotting, cellular ATP levels and ROS detection ? Your conclusion on the robustness of your results as a function of the number of samples is very welcome (l495-499). Our field of research limits us in the number of samples and it's always very relevant to point this out.
The number of samples is provided in the figure caption. However, it makes sense to also be explicit in the materials and methods sections. This correction was made.
For WB, “A total of 23 hCCs samples (12 from young group and 11 from AMA group) were recovered for the OXPHOS protein quantification.”
For ATP: “ ATP levels were measured by luciferin-luciferase assay in 33 samples, 24 from young group and 19 from AMA group”
For ROS: “ hCCs (39 samples, 21 young group and 19 AMA group) were incubated”
-l155 and l319 : Please be careful with the acronym « RT-qPCR ». According to the article « MIQE Guidelines : Minimum Information for Publication of Quantitative Real-Time PCR Experiments » (Bustin et al., 2009). « We propose that the abbreviation qPCR be used for quantitative real-time PCR and that RT-qPCR be used for reverse transcription– qPCR. Applying the abbreviation RT-PCR to qPCR causes confusion and is inconsistent with its use for conventional (legacy) reverse transcription–PCR. »
Thank you for your comment. The correction was made in the text.
- Can you explain the choice of relative mtDNA quantification?
The analysis of mtDNA content in absolute quantification is more appropriate and accurate. Moreover, it is impossible to relate your results to other articles in the literature (Ogino et al. 2016, Desquiret-Dumas et al. 2017, Taugourdeau et al. 2019, Yang et al 2021, Liu et al. 2021) describing an absolute quantification of mtDNA in cumulus cells.
In preparation for the mtDNA assay extensive research was conducted to identify the most robust methods for quantification. Our main conclusion was that the results concerning the mtDNA quantification are not consensual and this is due to a different approach used to perform the assay.
In our work, we chose an approach which seemed appropriate and blasted the primer sequence against the human genome. We did not find evidence that primers would bind to nuclear mitochondrial sequences. Thank you for providing us with this review. We appreciate the information and in the future this knowledge will be carefully considered in our analyses.
I invite the authors to read the article by Li et al (Pharmacol Ther 2021, Emerging methods for and novel insights gained by absolute quantification of mitochondrial DNA copy number and its clinical applications). This review summarises the methods used for mtDNA copy number quantification and in particular the advantages of absolute quantification.
-Mitochondrial primers have been tested on Rho0 cells? This is essential to avoid pseudogenes amplification.
We do not have this information. We can only state, as we do, that these primers were used by other authors. This issue is perhaps less relevant given the absence of differences in our samples, correlating with the lack of difference in mitochondrial mass.
Reviewer 3 Report
Comments and Suggestions for Authors
In this paper, COCs from patients were collected and analysed a multitude of ways measuring mitochondrial-related parameters.
The samples were arbitrarily split into 2 groups depending whether they were derived from women less than 35 years of age or 35 or older, and the entire cohorts demographics were listed n=140, n=126 resp). Also numbers dropped to 101 in the <35yo for fertilized eggs (presuming others may have had eggs frozen—they should not be in the demographic table if this is the case- or should be listed separately). I have a big issue with this report, as many of the tests were derived from a subset of this population (eg mito mass, n=27,n=28, DNA content, n=13, n=22, OCR, n=3, n=3). There was only one figure in the main paper showing the individual samples, and this was quite telling, as the 28yo sample looked very similar to the 37 yo sample, thus this would suggest that the groups were split wrongly. Also how do we know if the samples in the young group were taken from the fertile of non-fertile population in any sub cohort– any small difference could have been due to fertility issues rather than age. If this paper were to be published graphs would be better presented as scatter plots with age (per year) on the Y axis (as opposed to the box plots) and if you were considering age only then the fertile population should be removed or kept as a separate group, not just in the supplementary.
The mitochondrial images are also unconvincing. Need to be much higher magnification to detect changes.
I do not believe that you have presented the evidence that allows you to conclude ‘In conclusion, age-related hCCs dysfunction is mainly linked to an impairment in mitochondrial function, specifically a decrease in ATP synthase function.’ Thus I think this paper, written as it is, should be rejected.
Also please see specific comments on the original manuscript attached

The manuscript was easy to understand however would be improved with an english editor, some obvious errors are commented on in the original manuscript (see attached).
Author Response
Thank you for your comments and thorough review.
First of all, we completely agree that our initial conclusion was overstated. We have heavily edited the text (including the title) to better reflect the findings. There is a correlation in ATP synthase levels with mitochondrial function, but this is not a functional link and more samples/assays are needed. We hope this is now acceptable
It is obviously very hard to conduct experiments with human samples in this context, and to determine what the best “control” is. In relation to the cohort of women from whom oocytes were frozen (fertility preservation or oocyte donation), we conducted both demographic analysis and mitochondrial function assessments. These assays were performed separately and subsequently compared with the young group (Supplementary Figure 1). No significant differences were observed between the two groups, except for the evident age discrepancy — the individuals undergoing fertility preservation procedures were notably younger. However, the parameters were similar to the “younger” group, and better than the AMA group. This lends some confidence that age is the main factor at play here.
Regarding data presentation, we agree that the majority of images are only representative. It is important to clarify that our selection process aimed to choose the most illustrative images to support our findings. For instance, images depicting ∆Ψm or ROS are representative of their respective groups. We have provided other images but there are two points to be made: 1- the probes have been shown to be specifically mitochondrial in several studies in many cell types; 2- fluorescence intensity differences are quite clear in our opinion and, more importantly, conclusions were derived from “blind” quantification and appropriate statistical analyses, ensuring the robustness and validity of our results.
In Western Blot data the advantage lies in the ability to present a larger number of samples simultaneously. Our intention was to maximize sample representation when loading samples of different ages onto the gel. We can confidently state that the samples were accurately allocated to their respective groups. However, it is essential to recognise the inherent challenge of working with human samples, particularly the intra-group variance, which may be influenced by many factors.
We have some reservations regarding the suggested graph presentation proposed by the reviewer. For example, scatter plot per age was used for the presentation of ∆Ψm, however for WB or OCR (where different targets were analysed) this type of presentation is not appropriate, in our opinion.
Round 2
Reviewer 1 Report
Comments and Suggestions for Authors
Correctionts are sufficient, manuscript is acceptable in this form.